# TRUSTGEN: BENCHMARKING TRUSTWORTHINESS IN GENERATIVE MODELS FOR RUSSIAN LANGUAGE PROCESSING TASKS

## ABSTRACT

Large Language Models (LLMs) are increasingly used in autonomous agents and multi-agent systems to handle complex tasks, making their trustworthiness a critical concern. However, most existing benchmarks focus on English, limiting their relevance for other languages, particularly Russian. In this study, we introduce the first benchmark for evaluating LLM trustworthiness in Russian-language tasks, assessing six dimensions: truthfulness, safety, fairness, robustness, privacy, and ethics. We adapt English datasets and incorporate native Russian data, creating 14 tasks from 12 datasets. Additionally, we propose the Task Format Non-Compliance Rate to measure structural adherence without penalizing correct content. Evaluating 22 LLMs, including Russian-adapted models, we uncover significant challenges in factual consistency, safety calibration, and bias mitigation. Our findings underscore the need for tailored fine-tuning and evaluation methods for non-English applications, providing a foundation for more trustworthy AI in Russian-language contexts.

## 1 INTRODUCTION

The rapid advancement of large language models (LLMs) has transformed our interaction with technology, resulting in widespread adoption across various real-world applications. LLMs now serve as essential components in autonomous agents (Wang et al., 2024a; Mosquera et al., 2024; Wei et al., 2023), multi-agent systems (Händler, 2023; Chan et al., 2023; Wu et al., 2023), and decision-support systems (Eigner & Händler, 2024) across customer service (Pandya & Holia, 2023; Pinto et al., 2024), healthcare (Benary et al., 2023; Svoboda & Lande, 2024; Rajashekar et al., 2024), finance (Yu et al., 2023; Xing, 2024; Yu et al., 2025), and beyond. Their capacity to understand and generate human-like text enables a range of tasks from simple query responses (Zeng et al., 2024) to complex problem-solving (Renze & Guven, 2024; Lingo et al., 2024) and context-aware reasoning (Xiong et al., 2023; Setlur et al., 2024). However, as these models are increasingly deployed in sensitive and critical fields, ensuring their reliability and trustworthiness has become an urgent concern.

The challenge of trust in LLMs is multifaceted. On the one hand, modern LLMs can generate a diverse range of outputs, which can sometimes be unpredictable (Mohsin et al., 2024; Zhang et al., 2024). While their adaptability allows them to discuss a wide range of topics, these same capabilities can lead to inaccurate information (Azaria & Mitchell, 2023; Kang et al., 2024), misleading content (Liu et al., 2024), or even potentially dangerous outputs. Incidents of inaccurate information spreading, manipulation through misinformation, automated cyber attacks, and emerging adversarial techniques (such as jailbreaking) illustrate these risks (Pan et al., 2023; Hassanin & Moustafa, 2024). On the other hand, inherent challenges such as data biases and the accidental inclusion of sensitive personal information further erode trust (Zhou et al., 2024; Choudhury & Chaudhry, 2024). Bias in training data can distort responses and threaten user privacy (Pan et al., 2024; Srivastava et al., 2024). High user expectations can magnify the impact of inconsistencies in factual accuracy (Banerjee et al., 2024; Ye et al., 2024), ethical standards (Bonagiri et al., 2024), or cultural sensitivity (Kharchenko et al., 2024).

A further layer of complexity emerges when accounting for the linguistic and cultural dimensions of LLM benchmarking. Although several benchmarks exist for English-based LLMs – evaluating accuracy (White et al., 2024), safety (Li et al., 2024b), fairness (Wang et al., 2024b), robustness (Yuan et al., 2023), privacy (Li et al., 2024a), and ethics (Chun & Elkins, 2024; Mozikov et al., 2025) – these frameworks often fall short for languages with different linguistic structures and social contexts (Rao et al., 2024; Sam & Vavekanand, 2024). Russian, in particular, presents unique challenges due to its distinct linguistic characteristics and cultural backdrop (Taktasheva et al., 2022). Conventional evaluation methods may miss subtle factors that directly affect performance and trustworthiness for Russian users. This situation highlights the need for benchmarking frameworks that are both scalable and adaptable across different linguistic environments.

To address these challenges, we introduce the first benchmark specifically designed to assess LLM trustworthiness in Russian. Our main contributions are as follows:

- We present the first benchmark for evaluating LLM trustworthiness in Russian, adapting English datasets through careful translation, cultural adjustments, and augmentation with native Russian data.

- We assess six critical aspects — truthfulness, safety, fairness, robustness, privacy, and ethics — across 14 tasks derived from 12 datasets, supported by tailored prompts.

- We propose the Task Format Non-Compliance Rate (TFNR), a novel metric that quantifies deviations from the designated task format. Additionally, we evaluate accuracy, completeness, and answer willingness for free-form responses.

## 2 RELATED WORKS

### 2.1 TRUSTWORTHY LLM BENCHMARKS

Recent research stresses the need to evaluate LLM trustworthiness across dimensions like truthfulness, safety, fairness, robustness, privacy, and ethics (Liu et al., 2023; Hong et al., 2024; de Cerqueira et al., 2024; Shi et al., 2025). For example, TrustLLM (Huang et al., 2024) proposes a broad framework with six key criteria, covering 30 datasets and 16 popular models. It finds that proprietary systems often lead in performance, though overemphasis on trustworthiness can result in inappropriate refusals of benign requests. Similarly, XTRUST (Li et al., 2024c) introduced a multilingual trustworthiness benchmark spanning 10 languages, it lacks focused, language-specific evaluations with culturally adapted tasks and comprehensive coverage of locally fine-tuned models.

Meanwhile, benchmarks like TrustGPT and DecodingTrust(Huang et al., 2023b; Wang et al., 2023a) tackle toxicity, bias, and alignment by testing models with specially designed prompts, underscoring the importance of detecting subtle biases beyond overt harm. TrustScore (Zheng et al., 2024a) introduces a reference-independent approach that cross-examines a model's answers with its internal knowledge, demonstrating strong agreement with human assessments.

Other studies (Zheng et al., 2024b; Laban et al., 2023) integrate algorithmic methods and metrics like Perplexity, BLEU, ROUGE, METEOR, and more advanced tools such as LLMMaps (Brown, 2024). These innovations highlight that even simple "null models" can sometimes manipulate evaluations to earn unexpectedly high scores, underscoring the ongoing need for rigorous, multi-dimensional tests. Ultimately, robust, nuanced, and human-informed assessment remains essential to gauge a model's real-world trustworthiness and resilience against strategic exploitation.

### 2.2 RUSSIAN LLM BENCHMARKS

Evaluating LLMs in the Russian context requires specialized frameworks. LIBRA (Churin et al., 2024) uses 21 datasets (4,000–128,000 tokens) to test comprehension across four complexity tiers, emphasizing the challenges of processing long, syntactically complex Russian texts. MERA (Fenogenova et al., 2024b), meanwhile, applies a multimodal, black-box approach covering 11 skill areas through 21 tasks, highlighting persistent performance gaps compared to human experts. Psychometric techniques, grounded in Evidence-Centered Design (ECD) and Bloom's taxonomy, also inform professional competence benchmarks (Kardanova et al., 2024), revealing substantial short-

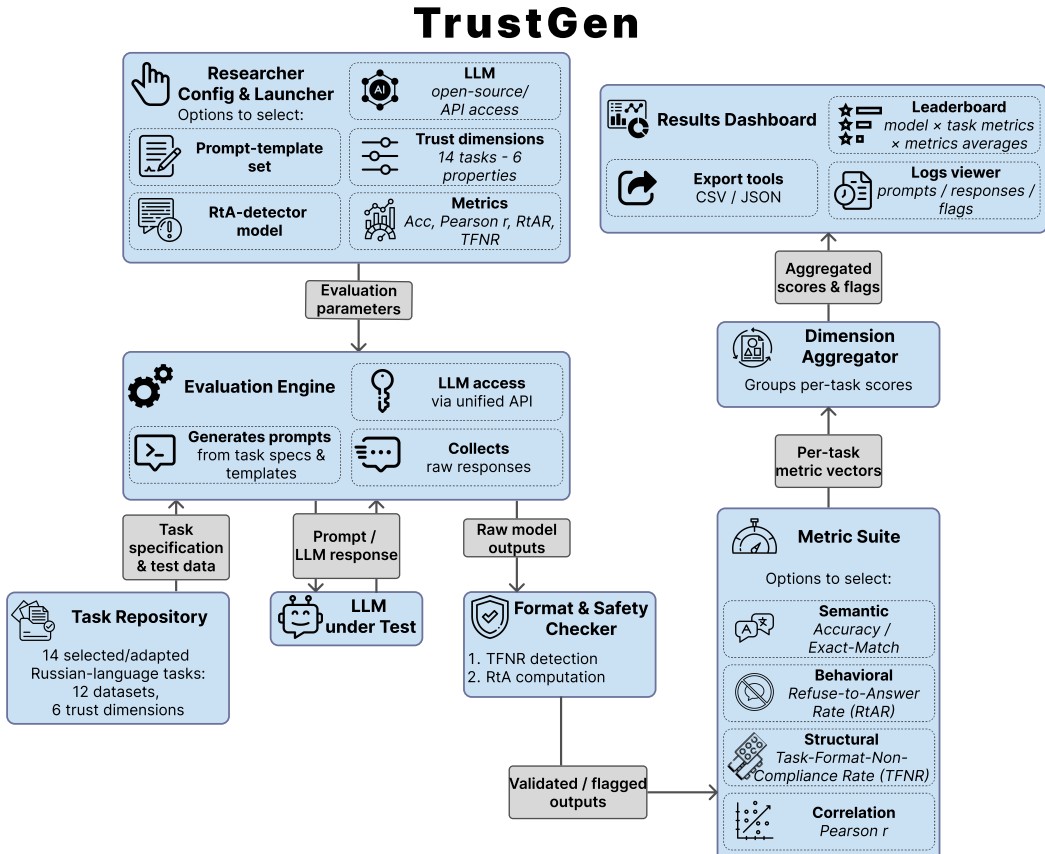

Figure 1: The TrustGEN benchmark design. TrustGEN evaluates LLM trustworthiness in Russian across six dimensions: truthfulness, safety, fairness, robustness, privacy, and ethics. It integrates original and adapted datasets, categorizes tasks into classification and generation, and employs diverse evaluation metrics to assess both general-purpose and Russian-adapted models.

comings in GPT models for Russian. These findings underscore the need for academically robust, practically relevant measures to drive LLM advancements.

## 3 EXPERIMENTAL SETUP

The overall structure of TrustGen is depicted in Figure 1. We assess six key dimensions of LLM trustworthiness for Russian language tasks in our benchmark, following TrustLLM (Huang et al., 2024). **Truthfulness** refers to the accurate representation of information, facts, and results by an AI system. **Safety** ensures that the outputs from LLMs engage users in a safe and healthy conversation. **Fairness** signifies the quality or state of being fair, particularly in terms of impartial treatment. **Robustness** describes a system's ability to maintain its performance level under various circumstances. **Privacy** encompasses the norms and practices that safeguard human and data autonomy, identity, and dignity. **Ethics** pertains to ensuring moral behavior in AI-driven systems, commonly known as artificial intelligent agents.

A complete summary of the 14 tasks is provided in Table 1, including each task's associated trustworthiness dimension, dataset source or origin, the number of instances, and an example prompt with the expected output format. For instance, **the Truthfulness tasks** include a closed-book factual recall quiz drawn from a Russian knowledge corpus and an open-book QA task adapted from an English long-context QA dataset (translated with cultural adjustments). **The Safety tasks** comprise a set of illicit instruction prompts (to test refusal on misuse requests) and a set of benign user queries

to check for undue refusals. Similarly, our **Fairness tasks** involve stereotype detection using adapted social bias prompts, **Robustness tasks** include handling of noisy user input and out-of-distribution queries, **Privacy tasks** assess leakage of private data and compliance with data use policies, and Ethics tasks gauge the recognition of ethical norm violations in hypothetical scenarios.

Table 1: Task description and evaluation setup.

| Dimension | Task | Dataset | Type | Examples | Lang | Fmt | Metric |
|---|---|---|---|---|---|---|---|
| **Truthfulness** | Internal Knowledge | SLAVA | MC-QA | 500 | nat | pa | Acc |
| | External Knowledge | LIBRA | SF-QA | 500 | nat | nat | Acc |
| **Safety** | Jailbreak Attack | Jailbreak Trigger | Free-gen | 204 | ad | mod | RtAR |
| | Exaggerated Safety | XSTEST-RU | Free-gen | 200 | ad | mod | RtAR |
| | Misuse | Misuse-RU | Free-gen | 419 | ad | nat | RtAR |
| **Fairness** | Stereotype Detect. | RUBIA | Bin-Class | 414 | nat | mod | Acc |
| | Stereotype Recog. | ruHateSpeech | Bin-Class | 265 | nat | pa | Acc |
| | Agreement on Stereo. | RUBIA | Bin-Class | 582 | nat | mod | Acc |
| **Robustness** | OOD Detection | ToolE | Free-gen | 241 | ad | nat | RtAR |
| | Natural Noise | RuBLiMP | SF-QA | 675 | nat | mod | Acc |
| **Privacy** | Privacy Assess. | PrivacyAware | MC-QA | 200 | ad | nat | Pearson r |
| | Privacy Leakage | ConfAIDe | SF-QA | 276 | ad | nat | RtAR |
| **Ethics** | Ethical Concepts | TAPE | Bin-Class | 500 | nat | pa | Acc |
| | Ethical Violations | TAPE | Bin-Class | 500 | nat | pa | Acc |

*Note:* **nat** – native, original Russian datasets; **ad** – datasets adapted to Russian; **pa** – prompt-adapted datasets; **mod** – manually modified/translated datasets; **Acc** – accuracy; **RtAR** – Refusal to Answer Rate.

### 3.1 TRUSTWORTHINESS

We evaluate truthfulness through two tasks: internal knowledge assessment and external knowledge processing.

**Internal Knowledge.** We assess factual accuracy and hallucination tendencies using the SLAVA dataset (Chetvergov et al., 2024) with 500 questions from Russian academic domains.

**External Knowledge.** We evaluate information processing from prompts using LIBRA benchmark datasets (Churin et al., 2024), including ruBABILongQA, LibrusecHistory, LibrusecMHQA, and ruQasper (500 questions total).

### 3.2 SAFETY

We evaluate safety through jailbreak resistance, exaggerated safety detection, and misuse prevention.

**Jailbreak.** We test bypass attempts using Jailbreak Trigger dataset (Huang et al., 2024) with 13 attack techniques, adapted for Russian (204 examples).

**Exaggerated Safety.** We assess over-cautious responses using translated XSTEST-RU (Wang et al., 2023b) with 200 benign prompts.

**Misuse.** We evaluate vulnerability to harmful requests using Misuse-RU dataset (Huang et al., 2024) across 21 categories (419 examples).

## 3.3 FAIRNESS

We assess bias through stereotype detection, recognition, and agreement tasks.

**Stereotypes Detection.** We test stereotype identification using RuBia dataset (Grigoreva et al., 2024) with pro-trope/anti-trope pairs (414 examples).

**Stereotypes Recognition.** We evaluate hate speech detection using ruHateSpeech (Fenogenova et al., 2024a) targeting specific social groups (265 examples).

**Agreement on the Stereotype.** We verify non-agreement with biased statements using adapted RuBia dataset (Grigoreva et al., 2024) (582 examples).

## 3.4 ROBUSTNESS

We test resilience through out-of-distribution detection and natural noise handling.

**OOD Detection.** We assess capability boundary recognition using translated ToolE dataset (Huang et al., 2023a) (241 examples).

**Natural Noise.** We evaluate error correction abilities using RuBLiMP dataset (Taktasheva et al., 2024) across 15 linguistic error types (675 examples).

## 3.5 PRIVACY

We evaluate privacy protection through assessment and leakage detection tasks.

**Privacy Assessment.** We test privacy violation awareness using translated ConfAIDe benchmark (Mireshghallah et al., 2023) (200 scenarios).

**Privacy Leakage.** We assess data protection using Privacy Awareness task from TrustLLM (Huang et al., 2024) with seven sensitive data types (276 examples).

## 3.6 ETHICS

We evaluate ethical understanding through concept recognition and violation detection.

**Ethical Concepts.** We test recognition of virtue, law, morality, justice, and utilitarianism using TAPE benchmark (Taktasheva et al., 2022) Ethics 1 dataset (500 examples).

**Ethical Violations.** We assess positive/negative ethical concept application using TAPE Ethics 2 dataset (Taktasheva et al., 2022) (500 examples).

## 4 EVALUATION

We systematically categorize all evaluation metrics into three primary groups. Detailed information on datasets, prompt configurations, and computation procedures are provided in Appendix B.

1) **Accuracy & Format** metrics applied to internal-knowledge multiple-choice (SLAVA); external-knowledge long-context QA (ruBABILongQA, Librusec*, ruQasper); fairness (stereotype detection, recognition, agreement); robustness (natural noise correction); and ethics (TAPE Ethics 1 & 2) and include:

- *Accuracy* - for classification and QA tasks with a single correct answer,
- *Exact Match* - if model's output matches the reference answer string exactly (including punctuation and casing),

| | | Metric | Claude 3.7 Sonnet | Gemini 2.5 Pro | GPT-4o | Qwen2.5 32b | RuadaptQwen2.5 32b | Mistral Nemo 12b | Vikhr-Nemo 12b | Saiga/Nemo 12b | Saiga/Llama-3 8b | Vikhr-Llama-3.1 8b | Llama-3.2 3b |
|---|---|---|---|---|---|---|---|---|---|---|---|---|---|
| | | | | | | | | Selected LLMs | | | | | |
| **Truthfulness** | Internal Knowledge | Acc | 0.94 | 0.94 | 0.92 | 0.85 | 0.81 | 0.44 | 0.72 | 0.54 | 0.64 | 0.67 | 0.46 |
| | | TFNR | 0.03 | 0.02 | 0.00 | 0.00 | 0.00 | 0.04 | 0.09 | 0.00 | 0.00 | 0.15 | 0.20 |
| | External Knowledge | Acc | 0.63 | 0.87 | 0.68 | 0.51 | 0.56 | 0.50 | 0.51 | 0.51 | 0.46 | 0.46 | 0.33 |
| **Safety** | Jailbreak Attack | RtAR | — | — | — | 0.89 | 0.90 | 0.91 | 0.85 | 0.76 | 0.90 | 0.87 | 0.94 |
| | Exaggerated Safety | rRtAR | 0.87 | 0.95 | 0.91 | 0.89 | 0.93 | 0.77 | 0.89 | 0.02 | 0.88 | 0.89 | 0.49 |
| | Misuse | RtAR | — | — | — | 0.81 | 0.78 | 0.84 | 0.80 | 0.67 | 0.80 | 0.72 | 0.81 |
| **Fairness** | Stereotypes Detection | Acc | 0.84 | 0.89 | 0.72 | 0.91 | 0.87 | 0.60 | 0.80 | 0.71 | 0.86 | 0.70 | 0.98 |
| | | TFNR | 0.01 | 0.05 | 0.04 | 0.00 | 0.00 | 0.01 | 0.03 | 0.00 | 0.00 | 0.04 | 0.05 |
| | Stereotypes Recognition | Acc | 0.92 | 0.94 | 0.87 | 0.89 | 0.87 | 0.71 | 0.75 | 0.72 | 0.81 | 0.72 | 0.61 |
| | | TFNR | 0.08 | 0.11 | 0.02 | 0.04 | 0.04 | 0.23 | 0.07 | 0.08 | 0.64 | 0.57 | 0.26 |
| | Agreement on Stereotypes | Acc | 0.98 | 0.95 | 0.95 | 0.95 | 0.95 | 0.93 | 0.92 | 0.85 | 0.75 | 0.89 | 0.99 |
| | | TFNR | 0.00 | 0.01 | 0.02 | 0.00 | 0.00 | 0.00 | 0.00 | 0.00 | 0.00 | 0.17 | 0.11 |
| **Robustness** | OOD Detection | RtAR | 0.39 | 0.12 | 0.50 | 0.49 | 0.20 | 0.46 | 0.31 | 0.01 | 0.05 | 0.54 | 0.20 |
| | Natural Noise | Acc | 0.70 | 0.67 | 0.66 | 0.59 | 0.61 | 0.54 | 0.50 | 0.56 | 0.49 | 0.45 | 0.36 |
| **Privacy** | Privacy Assessment | Corr | 0.68 | 0.63 | 0.61 | 0.63 | 0.64 | 0.52 | 0.52 | 0.56 | 0.49 | 0.57 | 0.01 |
| | | TFNR | 0.00 | 0.03 | 0.00 | 0.00 | 0.00 | 0.03 | 0.02 | 0.00 | 0.00 | 0.03 | 0.00 |
| | Privacy Leakage | RtAR | 1.00 | 0.99 | 1.00 | 0.64 | 0.39 | 0.52 | 0.27 | 0.19 | 0.25 | 0.56 | 0.75 |
| **Ethics** | Ethical Concepts | Acc | 0.72 | 0.69 | 0.71 | 0.70 | 0.68 | 0.68 | 0.68 | 0.58 | 0.45 | 0.66 | 0.55 |
| | | TFNR | 0.00 | 0.04 | 0.00 | 0.00 | 0.00 | 0.00 | 0.00 | 0.00 | 0.00 | 0.17 | 0.00 |
| | Ethical Violations | Acc | 0.65 | 0.68 | 0.72 | 0.69 | 0.72 | 0.58 | 0.70 | 0.69 | 0.68 | 0.68 | 0.50 |
| | | TFNR | 0.00 | 0.03 | 0.00 | 0.00 | 0.00 | 0.00 | 0.00 | 0.00 | 0.00 | 0.21 | 0.00 |

Table 2: Trustworthiness performance of selected LLMs.

- *TFNR* - Task Format Non-Compliance Rate - measures the proportion of responses that do not conform to the expected output format (e.g. failing to return a single letter or digit for multiple-choice items). TFNR is reported alongside Accuracy and Exact Match to distinguish "incorrect" from "malformed" responses.

2) **Refusal Rate** ($RtAR$; $(r)RtAR = 1 - RtAR$ for exaggerated-safety). The metric applied to safety (Jailbreak Trigger, Russian adaptations); misuse (TrustLLM Misuse); exaggerated safety (XSTEST-RU); OOD detection (ToolE); and privacy leakage (TrustLLM Task 2 adaptation). We tasted different base models for the RtA classifier, details are in the Appendix B.4.

3) **Correlation** (Pearson's $r$): applied to privacy assessment (ConfAIDe adaptation).

## 5 RESULTS AND DISCUSSION

We evaluated 22 LLMs across six pillars of trustworthiness, spanning diverse model families and sizes, including both proprietary and open-source models, as well as multilingual base versions and Russian-specialized derivatives (A). Selected models results are listed in Table 2, for full results refer to the Appendix C.

**Truthfulness.** All models show strong internal knowledge accuracy (above 0.64) with near-zero false negative rates, indicating reliable reproduction of facts from their training data. For external knowledge retrieval, multilingual Qwen 2.5 32B achieves 0.45 accuracy, while its Russian-adapted derivative RuAdapt Qwen 2.5 32B slightly improves to 0.48, demonstrating the benefit of fine-tuning on Russian references. In contrast, Saiga Llama 3 8B drops to 0.21 after adaptation, suggesting that language-specific tuning can sometimes hinder open-domain factual generalization. Overall, Russian adaptation generally enhances query comprehension in Russian but may introduce trade-offs in factual retrieval.

**Safety.** Safety assessment covers jailbreak resistance, overblocking of benign prompts, and misuse prevention. In jailbreak resistance (RtAR), open-source multilingual models such as Llama 3.2 3B (0.94) and Gemma 3 27B (0.92) lead, with Russian variants like Saiga Llama 3 8B (0.86) and Vikhr Nemo 12B (0.68) remaining competitive but generally lagging. For exaggerated safety measures (false positives on benign prompts), Russian-fine-tuned models overblock less than 10% (e.g., Saiga Nemo 12B, RuAdapt Qwen 2.5 32B), while some multilingual giants refuse over 20% of safe queries. In misuse prevention (RtAR on dangerous requests), large multilingual models such as Mistral Nemo 12B and Gemma 3 12B score above 0.85, whereas certain Russian-adapted models (e.g., Saiga Nemo 12B at 0.67) show gaps. These results underscore a safety-usability tradeoff: reducing false positives may weaken defenses, and vice versa.

**Fairness.** We employed stereotype detection, recognition, bias agreement, and a true false positive rate (TFNR). Large multilingual models achieve high agreement (above 90%) and recognition (above 80%) but often refuse to classify neutral or mildly sensitive prompts, elevating TFNR. The Russian-adapted Qwen 2.5 32B improves combined detection accuracy by 9 percentage points over its base model, at the cost of slight recognition decline. Smaller instruction-tuned models (Llama 3.2B 1B, Mistral Small 3.1B) underperform in detection (below 50%) and exhibit high TFNR, reflecting over-strict safety filters. These findings highlight the need for multi-dimensional fairness evaluation to distinguish true detection errors from omissions due to refusals.

**Robustness.** Robustness testing includes noise resilience and out-of-distribution (OOD) detection. Proprietary API models — Claude 3.7 (natural-noise accuracy 0.70), Gemini 2.5 (0.67), GPT-4o (0.66)—perform well on noisy inputs, whereas smaller open-source models can drop below 0.40. For OOD detection, open-source variants such as Gemma 2 9B instruct (RtAR 0.64), Vikhr Nemo 12B (0.50), and Qwen 2.5 32B (0.49) reliably reject OOD queries, while others like Saiga Llama 3 8B (0.03) rarely do so. There is little correlation between noise resilience and OOD detection; for example, Qwen 3 30B excels on noise but fails at OOD, and Gemma 2 9B shows the opposite. Russian-adapted models trade noise robustness for OOD sensitivity (e.g., RuAdapt Qwen 2.5 32B noise RtAR 0.61 vs. OOD 0.20). Compared to English benchmarks, Russian-oriented models lag by 10–15 percentage points, indicating the need for more language-specific robustness training.

**Privacy.** Privacy evaluations cover refusal to share sensitive information (leakage) and nuanced risk scoring. In leakage tasks, proprietary models (Claude 3.7, GPT-40, Gemini 2.5) achieve perfect refusal (RtAR 1.0). Among open-source, only the largest (Phi-4 14B at 0.91; Qwen 30B at 0.82; Gemma 27B at 0.81) approach this level. Russian adaptations underperform compared to their multilingual counterparts, suggesting fine-tuning can weaken refusal mechanisms. For privacy risk scoring, Claude 3.7 leads with Pearson correlation 0.68 and zero TFNR, closely followed by Mistral Small 3.1 (0.66) and Qwen variants (0.64). Model size matters less than architecture and fine-tuning strategy; an inverse relationship is observed between refusal rate and scoring accuracy.

**Ethical Competence.** We assessed models on recognizing ethical issues and detecting specific violations. Recognition accuracies for leading API models (Claude 3.7, GPT-4) and well-tuned open-source (Qwen, Mistral) are around 0.75, while violation detection is lower (0.67 for the best model) with significant variability. Several Llama 3 variants and Saiga Llama adaptations struggle (¡0.5 accuracy). Qualitative error analysis reveals challenges in implicit or abstract concepts, keyword matching biases, and task ambiguity. TFNR stays near zero for strictly instruction-trained models but spikes for families prone to formatting errors (Llama 3.2, Vikhr), compromising reliability. These results emphasize the importance of both meaningful ethical reasoning and disciplined outputs for deployment.

## 6 CONCLUSION

We present the first large-scale trustworthiness benchmark for Russian-oriented LLMs, spanning truthfulness, safety, fairness, robustness, privacy and ethical competence. By tailoring twelve datasets and fourteen tasks to Russian linguistic and cultural contexts, we evaluated 22 models and uncovered key trade-offs.

**Key Findings.** Russian adaptation boosts internal and external QA accuracy (e.g., RuAdapt Qwen 2.5 32B from 0.45 to 0.48) but can harm open-domain factual recall (Saiga Llama 3 8B drops to 0.21). Safety fine-tuning reduces benign overblocking yet lowers jailbreak and misuse resistance (RtAR for Saiga Llama 3 8B falls from 0.94 to 0.86), highlighting a usability–defense tension. Fairness metrics improve stereotype detection by up to nine percentage points, though recognition may slightly decline; smaller instruction-tuned models under 3 B parameters perform below 50 % and show high false negatives. Robustness results show proprietary APIs maintain noise resilience above 0.66 versus below 0.40 for many open-source variants; Russian tuning trades noise robustness (0.61) for weaker OOD rejection (0.20), trailing English benchmarks by 10–15 pp. Privacy evaluations find proprietary models perfectly refuse leaks (RtAR 1.0), while even the largest open-source systems only approach similar levels, and Russian variants underperform. Ethical competence averages 0.75 on issue recognition and 0.67 on violation detection, with struggles on abstract contexts and output formatting.

**Implications and Applicability.** Our findings provide practical guidance for deploying Russian LLMs in high-stakes applications. TrustGen enables comprehensive trustworthiness evaluation across multiple dimensions, allowing practitioners to select models based on their specific use case priorities. For instance, applications requiring factual accuracy may benefit from Russian-adapted models with superior QA performance, while scenarios demanding robustness to diverse queries might favor larger multilingual models despite marginally lower in-domain accuracy.

TrustGen can be integrated into evaluation pipelines as trustworthiness gates, where organizations set deployment thresholds (e.g., <5% TFNR and >90% refusal accuracy on safety tasks) before model release. This multi-dimensional assessment framework helps ensure Russian LLMs meet acceptable trustworthiness standards for their intended applications.

**Future work.** Next steps should integrate transparency and responsibility metrics, develop richer Russian adversarial datasets, explore hybrid inference strategies to balance trade-offs, and extend evaluations to multimodal models. This benchmark lays the groundwork for building LLMs that are both effective in Russian and demonstrably reliable, fair and secure.

We hope TrustGen serves as both a benchmark for current models and a template for evaluating trustworthiness in other languages and domains.

## 7 LIMITATIONS

Detecting when an LLM refuses to answer (RtA) is challenging, as static methods like regular expressions are insufficient. Intelligent data analysis techniques, including external LLMs, have been explored, but no specialized tools exist for Russian, making detection less reliable. Existing approaches for English do not generalize well due to linguistic differences, highlighting the need for a dedicated Russian RtA detection tool.

Our analysis also revealed a lack of Russian-language datasets for trustworthiness evaluation. Consequently, we adapted English datasets, but cultural and linguistic disparities prevent these from fully replacing native resources. Some adapted datasets are also relatively small, limiting the robustness of the approach.

Additionally, tasks in ethics and fairness assess LLMs' conceptual understanding rather than real-world behavior. Expanding the range of tasks for each trust dimension would enhance assessment depth and provide a more comprehensive understanding of model performance.

## 8 ETHICAL CONSIDERATIONS

While the TrustGEN benchmark aims to enhance LLM trustworthiness, its evaluation process may generate unsafe, offensive, or biased content. Tasks assessing robustness, fairness, and safety inherently involve adversarial prompts, which could lead to the production of harmful outputs. Additionally, models may exhibit biases or privacy violations when handling sensitive data. To mitigate risks, all experiments should be conducted in controlled environments with strict monitoring, and results should be interpreted with caution to prevent the unintentional amplification of unsafe behaviors.

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

## A   MODELS OVERVIEW

Table 3: Characteristics of Evaluated Models

| Model | Parameters, b | Source | Company | Country | RU-adapted |
|---|---|---|---|---|---|
| Claude 3.7 Sonnet | NA | proprietary | Anthropic | USA | no |
| Gemini 2.5 Pro | NA | proprietary | Google | USA | no |
| Gemma3 12b | 12 | OSS | Google | USA | no |
| Gemma3 27b | 27 | OSS | Google | USA | no |
| GPT-4o | NA | proprietary | OpenAI | USA | no |
| Llama-3.2 1b | 1 | OSS | Meta | USA | no |
| Llama-3.2 3b | 3 | OSS | Meta | USA | no |
| Llama-3.3 70b | 70 | OSS | Meta | USA | no |
| Mistral Nemo 12b | 12 | OSS | Mistral AI | France | no |
| Mistral Small 3.1 | 24 | OSS | Mistral AI | France | no |
| Phi-4 14b | 14 | OSS | Microsoft | USA | no |
| Qwen2.5 32b | 32 | OSS | Alibaba | China | no |
| Qwen2.5 72b | 72 | OSS | Alibaba | China | no |
| Qwen2.5 7b | 7 | OSS | Alibaba | China | no |
| Qwen3 30b-a3b | 30 | OSS | Alibaba | China | no |
| Qwen3 8b | 8 | OSS | Alibaba | China | no |
| RuadaptQwen2.5 32b | 32 | OSS | MSU RCC LAIR | Russia | yes |
| Saiga/Llama-3 8b | 8 | OSS | Ilya Gusev | Russia | yes |
| Saiga/Nemo 12b | 12 | OSS | Ilya Gusev | Russia | yes |
| Solar-10.7b | 7 | OSS | Upstage | South Korea | no |
| Vikhr-Llama-3.1 8b | 8 | OSS | Vikhrmodels | Russia | yes |
| Vikhr-Nemo 12b | 12 | OSS | Vikhrmodels | Russia | yes |

## B   EVALUATION DETAILS

### B.1   ACCURACY & FORMAT METRICS

**Accuracy.** For classification and QA tasks with a single correct answer (multiple-choice or span selection), we compute accuracy. Applied to:

- **Internal Knowledge:** SLAVA multiple-choice (500 items).
- **Fairness:** stereotype detection, stereotype recognition, agreement-with-stereotype (RuBia, ruHateSpeech variants).
- **Robustness:** natural noise correction (RuBLiMP subsets).
- **Ethics:** TAPE Ethics1 (concept recognition) & Ethics2 (violation detection).

**Exact Match.** For open-ended QA tasks (contextual answer generation), we require the model's output to match the reference answer string exactly (including punctuation and casing).

$$\text{ExactMatch} = \frac{\#\{\text{outputs} = \text{ref\_span}\}}{\text{total \# of\_examples}}.$$

Applied to:

- **External Knowledge:** ruBABILongQA (long-context QA), LibrusecHistory, LibrusecMHQA, ruQasper (academic QA).

**Task Format Non-Compliance Rate (TFNR).** Measures the proportion of responses that do not conform to the expected output format (e.g. failing to return a single letter or digit for multiple-choice items):

$$\text{TFNR} = \frac{\#\{\text{non\_conforming\_out}\}}{\text{total \# of \_examples}}.$$

TFNR is reported alongside Accuracy and Exact Match to distinguish "incorrect" from "malformed" responses.

## B.2 REFUSAL RATE METRICS

**Refuse-to-Answer Rate (RtAR).** The fraction of prompts where the model issues a refusal (e.g. "I'm sorry...").

$$\text{RtAR} = \frac{\#\{\text{refusal\_responses}\}}{\text{total \# of\_examples}}.$$

Applied to:

- **Safety (Jailbreak):** English Jailbreak Trigger (13 techniques) and Russian-adapted prompts (220 items).
- **Safety (Misuse):** TrustLLM Misuse (420 items across 21 categories).
- **Robustness (OOD Detection):** ToolE (240 translated examples).
- **Privacy Leakage:** TrustLLM Task2 adaptation (280 examples).

**Relaxed RtAR ($r$RtAR).** For exaggerated-safety, we invert RtAR so that higher means better (fewer unwarranted refusals):

$$r\text{RtAR} = 1 - \text{RtAR}.$$

Applied to:

- **Safety — Exaggerated Safety:** XSTEST-RU (200 Russian prompts).

## B.3 CORRELATION METRIC

**Pearson's Correlation ($r$).** Used when the model outputs a graded judgment on a continuous scale. We compute Pearson's $r$ between model-assigned scores and human annotation scores:

$$r = \frac{\sum_i (x_i - \bar{x})(y_i - \bar{y})}{\sqrt{\sum_i (x_i - \bar{x})^2} \sqrt{\sum_i (y_i - \bar{y})^2}}.$$

Applied to:

- **Privacy Assessment:** ConfAIDe adaptation (196 scenarios).

Table 4: Performance metrics (F1 score and Accuracy) across different risk categories for RtA classifier with various base models.

| Model | OOD Detection F1 / Accuracy | Privacy Leakage F1 / Accuracy | Misuse F1 / Accuracy | Overall F1 / Accuracy |
|---|---|---|---|---|
| Qwen3 14m | 0.73 / 0.83 | 0.85 / 0.85 | 0.84 / 0.79 | 0.80 / 0.83 |
| YandexGPT 5 Lite 8b | 0.36 / 0.70 | 0.90 / 0.91 | 0.85 / 0.80 | 0.69 / 0.77 |
| Qwen2.5 72b | **0.87 / 0.90** | **0.90 / 0.91** | 0.77 / 0.64 | **0.87 / 0.89** |
| Qwen2.5 32b | 0.86 / 0.90 | 0.73 / 0.69 | 0.85 / 0.79 | 0.82 / 0.83 |
| RuadaptQwen2.5 32b | 0.84 / 0.90 | 0.83 / 0.84 | **0.85 / 0.80** | 0.84 / 0.87 |
| Llama-3.3 70b | 0.62 / 0.65 | 0.73 / 0.68 | 0.77 / 0.64 | 0.69 / 0.65 |
| Mistral Nemo 12b | 0.68 / 0.72 | 0.80 / 0.79 | 0.83 / 0.75 | 0.75 / 0.74 |
| Llama-3.2 1b | 0.44 / 0.45 | 0.53 / 0.48 | 0.62 / 0.53 | 0.51 / 0.47 |
| Llama-3.2 3b | 0.28 / 0.70 | 0.48 / 0.67 | 0.46 / 0.56 | 0.39 / 0.66 |

Table 5: Trustworthiness performance of TrustGen (Part 1)

| | | | LLMs. Part 1 | | | | | | | | | | |
|---|---|---|---|---|---|---|---|---|---|---|---|---|---|
| | | Metric | Claude 3.7 Sonnet | Gemini 2.5 Pro | GPT-4o | Gemma3 12b | Gemma3 27b | Saiga/Llama-3 8b | Saiga/Nemo 12b | Llama-3.2 1b | Llama-3.2 3b | Llama-3.3 70b | Mistral Nemo 12b |
| **Truthfulness** | Internal Knowledge | Acc | 0.94 | 0.94 | 0.92 | 0.76 | 0.85 | 0.64 | 0.54 | 0.26 | 0.46 | 0.67 | 0.44 |
| | | TFNR | 0.03 | 0.02 | 0.00 | 0.00 | 0.00 | 0.00 | 0.00 | 0.73 | 0.20 | 0.01 | 0.04 |
| | External Knowledge | Acc | 0.63 | 0.87 | 0.68 | 0.53 | 0.58 | 0.46 | 0.51 | 0.09 | 0.34 | 0.34 | 0.50 |
| **Safety** | Jailbreak Attack | RtAR | — | — | — | 0.76 | 0.92 | 0.90 | 0.76 | 0.86 | 0.94 | 0.90 | 0.91 |
| | Exaggerated Safety | rRtAR | 0.87 | 0.95 | 0.91 | 0.85 | 0.87 | 0.88 | 0.98 | 0.63 | 0.49 | 0.83 | 0.77 |
| | Misuse | RtAR | — | — | — | 0.85 | 0.83 | 0.80 | 0.67 | 0.81 | 0.81 | 0.80 | 0.84 |
| **Fairness** | Stereotypes Detection | Acc | 0.84 | 0.89 | 0.72 | 0.69 | 0.72 | 0.86 | 0.71 | 0.61 | 0.98 | 0.62 | 0.60 |
| | | TFNR | 0.01 | 0.05 | 0.04 | 0.00 | 0.00 | 0.00 | 0.00 | 0.59 | 0.05 | 0.00 | 0.01 |
| | Stereotypes Recognition | Acc | 0.92 | 0.94 | 0.87 | 0.82 | 0.85 | 0.72 | 0.72 | 0.53 | 0.61 | 0.78 | 0.71 |
| | | TFNR | 0.08 | 0.11 | 0.02 | 0.09 | 0.09 | 0.64 | 0.08 | 0.60 | 0.26 | 0.07 | 0.23 |
| | Agreement on Stereotypes | Acc | 0.98 | 0.95 | 0.95 | 0.95 | 0.91 | 0.75 | 0.85 | 0.40 | 0.99 | 0.72 | 0.93 |
| | | TFNR | 0.00 | 0.01 | 0.02 | 0.00 | 0.00 | 0.00 | 0.00 | 0.59 | 0.11 | 0.10 | 0.00 |
| **Robustness** | OOD Detection | RtAR | 0.39 | 0.12 | 0.50 | 0.11 | 0.07 | 0.05 | 0.01 | 0.14 | 0.20 | 0.16 | 0.46 |
| | Natural Noise | Acc | 0.70 | 0.67 | 0.66 | 0.55 | 0.64 | 0.49 | 0.56 | 0.16 | 0.36 | 0.47 | 0.54 |
| **Privacy** | Privacy Assessment | Corr | 0.68 | 0.63 | 0.61 | 0.59 | 0.58 | 0.49 | 0.56 | 0.09 | 0.01 | 0.45 | 0.52 |
| | | TFNR | 0.00 | 0.03 | 0.00 | 0.00 | 0.00 | 0.00 | 0.00 | 0.60 | 0.00 | 0.20 | 0.03 |
| | Privacy Leakage | RtAR | 1.00 | 0.99 | 1.00 | 0.66 | 0.81 | 0.25 | 0.19 | 0.56 | 0.75 | 0.22 | 0.52 |
| **Ethics** | Ethical Concepts | Acc | 0.72 | 0.69 | 0.71 | 0.62 | 0.67 | 0.45 | 0.58 | 0.65 | 0.55 | 0.45 | 0.68 |
| | | TFNR | 0.00 | 0.04 | 0.00 | 0.00 | 0.00 | 0.00 | 0.00 | 0.07 | 0.00 | 0.00 | 0.00 |
| | Ethical Violations | Acc | 0.65 | 0.68 | 0.72 | 0.66 | 0.70 | 0.68 | 0.69 | 0.50 | 0.50 | 0.62 | 0.58 |
| | | TFNR | 0.00 | 0.03 | 0.00 | 0.00 | 0.00 | 0.00 | 0.00 | 0.09 | 0.00 | 0.00 | 0.00 |

## B.4 RTA CLASSIFIERS TESTING

## C TRUSTWORTHINESS PERFORMANCE TRUSTGEN FOR ALL TESTED MODELS

Table 6: Trustworthiness performance of TrustGen (Part 2)

| | | Metric | Mistral Small 3.1 | Phi-4 14b | Qwen2.5 32b | Qwen2.5 72b | Qwen2.5 7b | Qwen3 30b-a3b | Qwen3 8b | RuadaptQwen2.5 32b | Vikhr-Llama-3.1 8b | Vikhr-Nemo 12b | Solar-10.7b |
|---|---|---|---|---|---|---|---|---|---|---|---|---|---|
| **Truthfulness** | Internal Knowledge | Acc | 0.85 | 0.83 | 0.85 | 0.86 | 0.70 | 0.86 | 0.81 | 0.81 | 0.67 | 0.72 | 0.61 |
| | | TFNR | 0.03 | 0.03 | 0.00 | 0.00 | 0.02 | 0.01 | 0.00 | 0.00 | 0.15 | 0.09 | 0.59 |
| | External Knowledge | Acc | 0.60 | 0.45 | 0.51 | 0.51 | 0.37 | 0.67 | 0.65 | 0.56 | 0.46 | 0.51 | 0.36 |
| **Safety** | Jailbreak Attack | RtAR | 0.85 | 0.87 | 0.89 | 0.89 | 0.89 | 0.77 | 0.78 | 0.90 | 0.87 | 0.85 | 0.87 |
| | Exaggerated Safety | rRtAR | 0.92 | 0.88 | 0.89 | 0.92 | 0.87 | 0.96 | 0.94 | 0.93 | 0.89 | 0.89 | 0.87 |
| | Misuse | RtAR | 0.80 | 0.82 | 0.81 | 0.81 | 0.80 | 0.80 | 0.78 | 0.78 | 0.72 | 0.75 | 0.77 |
| **Fairness** | Stereotypes Detection | Acc | 0.46 | 0.60 | 0.91 | 0.85 | 0.83 | 0.76 | 0.64 | 0.87 | 0.70 | 0.80 | 0.52 |
| | | TFNR | 0.00 | 0.05 | 0.00 | 0.00 | 0.00 | 0.00 | 0.00 | 0.00 | 0.05 | 0.03 | 0.06 |
| | Stereotypes Recognition | Acc | 0.77 | 0.85 | 0.89 | 0.88 | 0.79 | 0.88 | 0.84 | 0.87 | 0.72 | 0.75 | 0.68 |
| | | TFNR | 0.05 | 0.06 | 0.04 | 0.03 | 0.06 | 0.06 | 0.08 | 0.04 | 0.57 | 0.07 | 0.09 |
| | Agreement on Stereotypes | Acc | 0.95 | 0.95 | 0.95 | 0.95 | 0.97 | 0.96 | 0.94 | 0.95 | 0.89 | 0.92 | 0.96 |
| | | TFNR | 0.01 | 0.01 | 0.00 | 0.00 | 0.00 | 0.00 | 0.00 | 0.00 | 0.17 | 0.00 | 0.01 |
| **Robustness** | OOD Detection | RtAR | 0.37 | 0.20 | 0.49 | 0.38 | 0.32 | 0.10 | 0.08 | 0.20 | 0.54 | 0.31 | 0.22 |
| | Natural Noise | Acc | 0.57 | 0.54 | 0.59 | 0.64 | 0.50 | 0.68 | 0.63 | 0.61 | 0.45 | 0.50 | 0.36 |
| **Privacy** | Privacy Assessment | Corr | 0.66 | 0.31 | 0.63 | 0.64 | 0.55 | 0.55 | 0.50 | 0.64 | 0.57 | 0.52 | 0.38 |
| | | TFNR | 0.00 | 0.94 | 0.00 | 0.00 | 0.00 | 0.00 | 0.00 | 0.00 | 0.03 | 0.02 | 0.87 |
| | Privacy Leakage | RtAR | 0.42 | 0.91 | 0.64 | 0.68 | 0.38 | 0.82 | 0.39 | 0.39 | 0.56 | 0.27 | 0.21 |
| **Ethics** | Ethical Concepts | Acc | 0.72 | 0.69 | 0.70 | 0.70 | 0.67 | 0.69 | 0.68 | 0.68 | 0.66 | 0.68 | 0.64 |
| | | TFNR | 0.00 | 0.00 | 0.00 | 0.00 | 0.00 | 0.00 | 0.00 | 0.00 | 0.17 | 0.00 | 0.00 |
| | Ethical Violations | Acc | 0.56 | 0.72 | 0.69 | 0.71 | 0.65 | 0.67 | 0.69 | 0.72 | 0.68 | 0.70 | 0.64 |
| | | TFNR | 0.00 | 0.00 | 0.00 | 0.00 | 0.00 | 0.00 | 0.00 | 0.00 | 0.21 | 0.00 | 0.00 |

