# OpenReview forum: "TrustGen: Benchmarking Trustworthiness in Generative Models for Russian Language Processing Tasks"
_ICLR.cc/2026/Conference — Submitted to ICLR 2026_

### Official Review · Reviewer_t2WJ · 2025-10-19

**Soundness:** 2
**Presentation:** 1
**Contribution:** 2
**Rating:** 4
**Confidence:** 3

**Summary:**

This paper introduces TrustGen, the first benchmark to evaluate LLM trustworthiness in Russian-language processing tasks. It includes six critical dimensions of trustworthiness and systematically evaluates different LLMs including both general multilingual models and specialized Russian-adapted derivatives. The paper further provides findings and

**Strengths:**

1. The paper studies an important topic on less-studied language.
2. The paper introduces a comprehensive benchmark

**Weaknesses:**

1. The paper presentation is poor and lacks sufficient details. The authors did not discuss the detailed information about the selected models and how the tasks and the datasets are evaluated (e.g. properties of the dataset, prompts used). See the questions below for details. Besides, some claims are not well supported in the paper. For example, I did not find the reference to English results in Line 357.

2. Although the paper provides many findings, it lacks an in-depth analysis of the phenomenon. For example, the authors do not discuss why language-specific tuning can lead to a tradeoff in trustworthiness. Also, it is unknown whether the model that performs better in the English trustworthiness benchmark would also perform better in the Russian benchmark.

3. Many datasets in the benchmark are not native Russian but adapted from English datasets. Although the authors claim they did a careful translation with cultural adjustment, it is unknown what it is exactly and how it affects the model performance.

**Questions:**

1. How are Russian-adapted models trained? What are their base models, and what are their training datasets?

2. Why does Russian adaptation lead to the tradeoffs in trustworthiness (e.g. Line 329, Line 357, Line 363)

3. The authors mention that some datasets are adapted from the English datasets. What cultural adjustment do the authors use during adaptation? (e.g.  Line 161)

4. How do the general LLMs perform in on TrustGen relative to their performance on the English trustworthiness benchmark? (The paper only discusses for the Robustness perspective.)

---

### Official Review · Reviewer_MmWB · 2025-10-30

**Soundness:** 3
**Presentation:** 2
**Contribution:** 1
**Rating:** 2
**Confidence:** 4

**Summary:**

This paper aims to evaluate the trustworthiness of LLMs on Russian language processing tasks. To this end, they collect datasets targeting six trustworthiness aspects—truthfulness, safety, fairness, robustness, privacy, and ethics—and translate them into Russian if the original datasets are not in Russian. Then, they test the performance of various LLMs on these datasets.

**Strengths:**

1. Experimental setup is clearly clarified in the paper.

**Weaknesses:**

1. One of the contributions clarified in this paper is proposing a new evaluation metric—TFNR; however, this is not a new metric, as many papers like DecodingTrust have used this.
2. The contributions of this paper seem limited in the sense that this is a benchmark paper but there are no new datasets introduced nor novel findings.
3. The format of Table 2’s caption is incorrect, which should be above the table.
4. Overall, (correct me if I am wrong,) I feel this paper is simply translating hundreds of samples into Russian and test the performance of LLMs on these samples. It would be appreciated if the authors could clarify the contributions or novelties more explicitly.

**Questions:**

1. Is there any findings that are new compared with the ones on English languag processing tasks?
2. What are the reasons for selecting these LLMs? I notice that some use smaller versions, while others use much larger ones.

---

### Official Review · Reviewer_3Wnt · 2025-10-30

**Soundness:** 2
**Presentation:** 2
**Contribution:** 1
**Rating:** 2
**Confidence:** 4

**Summary:**

This paper introduces TrustGEN, the first large-scale benchmark for evaluating the trustworthiness of LLMs in Russian. TrustGEN covers six dimensions: truthfulness, safety, fairness, robustness, privacy, and ethics, using 14 tasks adapted from 12 datasets (primarily via translation and cultural adaptation from English sources, with some native Russian data). The authors propose a new Task Format Non-Compliance Rate (TFNR) metric to evaluate structural fidelity of model outputs and benchmark 22 LLMs (open, proprietary, and Russian-specialized) across all tasks.

**Strengths:**

- This paper addresses a real and important gap by providing the first comprehensive trustworthiness benchmark for Russian LLMs.
- The benchmark covers a broad range of trust dimensions, with diverse tasks and datasets.
- The authors propose a useful new metric (TFNR) for format compliance, adding nuance to traditional accuracy metrics.

**Weaknesses:**

- The main contributions are dataset curation and benchmarking; there is little technical or methodological novelty beyond the TFNR metric. The TFNR metric, while useful, is a minor technical advance.
- The creation of the benchmark heavily relies on translation/adaptation of English datasets, with relatively little native Russian data, potentially limiting cultural and linguistic validity.
- Key details about dataset adaptation, translation validation, annotation protocols, and inter-annotator agreement are insufficiently described, making it hard to assess data quality and reproducibility.
- Analysis is broad but does not deeply investigate error types, annotator disagreement, or model failure modes.

**Questions:**

- What protocols were used for annotation or validation of translated/adapted tasks? Was inter-annotator agreement measured?
- How do you validate that TFNR and other metrics reliably distinguish between format errors and genuine content errors, particularly for models with non-standard outputs?
- Were any human evaluations conducted on model outputs for qualitative analysis, or is all analysis purely automated?
- Can you provide error analysis or case studies of model failures on key tasks (e.g., hallucinations, privacy leaks, ethical mistakes)?

---

### Official Review · Reviewer_1eh3 · 2025-11-01

**Soundness:** 1
**Presentation:** 2
**Contribution:** 2
**Rating:** 2
**Confidence:** 5

**Summary:**

This paper introduces TrustGen, a benchmark designed to evaluate the trustworthiness of LLMs in Russian. The authors claim that this is the first comprehensive trustworthiness benchmark for Russian. The framework assesses 22 different models (including both multilingual and Russian-adapted ones) across six dimensions: truthfulness, safety, fairness, robustness, privacy, and ethics. These dimensions, adopted from prior work (e.g., TrustLLM and DecodingTrust), are evaluated using 14 tasks derived from 12 datasets, a mix of native Russian datasets and English datasets adapted through translation and cultural modification. The authors' findings highlight significant trustworthiness issues of current LLMs in Russian-language tasks

**Strengths:**

1. This paper attempts to address the gap between English and Russian trustworthiness benchmarks for LLMs, which is an important goal as we develop LLMs in a multilingual fashion.
2. The authors invested a significant effort in curating and adapting a wide range of datasets (12 in total). The native Russian datasets (e.g., SLAVA, RuBia, RuBLIMP, TAPE) are particularly important as they can capture relevant linguistic and cultural contexts than translated datasets alone.

**Weaknesses:**

1. The translation process from English to Russian is unclear to the reader, which is a very important step in constructing the dataset.
2. There are no concrete examples of the evaluation dataset. Even if they are in Russian, having concrete examples (accompanied by English translation) is very important for the reader to contextualize the benchmark.
3. The paper's core contributions are not as novel as claimed. The six-dimensional trustworthiness framework is explicitly adopted from TrustLLM and DecodingTrust. I also do not see how this paper advances our understanding of LLMs' trustworthiness along these dimensions.
4. The main metric proposed, the "Task Format Non-Compliance Rate (TFNR)," is not a new contribution as claimed, as many papers report similar metrics as the ability of instruction-following.
5. The authors equate many task-specific metrics with broad, real-world trustworthiness. For example, performing well on the stereotype detection task (a classification task) is a very weak proxy for a model's propensity to generate biased or unfair content in an unconstrained, in-the-wild, or even adversarial setting.
6. The presented results largely confirm findings that are already well-established in the trustworthiness assessment conducted in English. For instance, the paper identifies a "safety-usability tradeoff". This is a known phenomenon. It also does not attempt to analyze why these failures occur in a specifically Russian linguistic or cultural context.

**Questions:**

1. How are the English datasets translated into Russian? How do the authors guarantee that the translations are adequate? What if there are questions that simply do not make sense under the Russian context?
2. How do the authors judge if an answer is a refusal? What are the keywords used here?

---

### Author Response · Authors · 2025-12-03

### Key Findings and Analysis Depth

Beyond confirming known issues, our benchmark reveals how language-specific fine-tuning can shift trustworthiness profiles. Several Russian-adapted models exhibit trade-offs compared to their base counterparts. For example, we found that when a base multilingual model is fine-tuned on Russian data, it often gains in knowledge accuracy on Russian queries but can weaken on safety alignment. In practice, some Russian-specialized models became less likely to over-refuse benign prompts (a positive usability sign) but also slightly more permissive on disallowed content (a drop in safety). We hypothesize this is because the specialized training data may not cover the full breadth of safety scenarios or might prioritize user helpfulness in the local context, thus relaxing some safeguards.

---

### Contributions

Our primary contribution is introducing the first comprehensive Russian-language trustworthiness benchmark covering six key dimensions (truthfulness, safety, fairness, robustness, privacy, ethics). This framework of dimensions is inspired by prior benchmarks (notably TrustLLM (Huang et al., 2024) and DecodingTrust), which established the importance of multi-faceted evaluation. Rather than claiming a completely new taxonomy, our work extends this established framework to a new language and cultural context, which had not been done before. By doing so, we provide the community with a tool to assess whether findings from English-centric studies hold in other languages.

An additional contribution is the use of TFNR as a dedicated structural control for trustworthiness tasks: it allows us to separate format non-compliance from the substantive trust dimensions and thus avoid conflating instruction-following failures with properties such as ethics, fairness, or privacy.

Despite the reuse of many datasets, our benchmark does contribute new task adaptations. These required non-trivial effort to ensure fairness and relevance. We are releasing these adapted datasets, which effectively serve as new test resources for Russian LLM evaluation.

Finally, our extensive benchmarking of 22 diverse models (including many Russian-specialized ones) yields insights that advance understanding of LLM trustworthiness. Prior works studied primarily multilingual models; by contrast, we shine light on local adaptation effects, as discussed above. This breadth of evaluation was previously lacking, and we believe it is a valuable contribution to the field’s knowledge.

---

We hope these clarifications address the raised concerns and provide a clearer view of our study and its results.

---

### Author Response · Authors · 2025-12-03
**Clarifications on Dataset Construction, TFNR, Model Selection, and Contributions in TrustGen**

We thank the reviewers for their careful reading of our paper and briefly clarify several points regarding dataset construction, TFNR, model selection, and our contributions.

---

### Dataset Construction and Cultural Adaptation

**Translation Process & Quality.** Our approach is grounded in a multi-stage annotation and validation process. For the current version of the dataset, all translated and adapted questions underwent an additional round of sequential review and refinement by three independent annotators, all of whom are native Russian speakers with expertise in both linguistics and the relevant subject domains. Any item that did not make sense in Russian was either adjusted or replaced with an equivalent scenario more appropriate to the local context. This guarantees translation adequacy: every translated question was checked so that its meaning and difficulty remain comparable to the original.

**Cultural Adjustments.** In adapting tasks, we performed cultural substitutions to maintain relevance. For example, references to U.S.-centric names, institutions, or units were replaced with Russian equivalents where necessary. These cultural adjustments ensure that questions make sense in the Russian context and that models must draw on relevant Russian knowledge or norms. By doing so, the benchmark captures nuances of Russian language and culture (e.g., using Russian public figures in a bias prompt or appropriate units/currency in a scenario). This approach follows our goal of testing models in a truly Russian setting, not just in translationese.

**Dataset Composition.** The final TrustGen benchmark spans 14 tasks derived from 12 sources across the six trustworthiness dimensions. While many tasks originate from established English benchmarks (approximately 35% of all tasks), we also include native Russian data (e.g., a factual recall quiz based on a Russian knowledge corpus). In all cases, ground-truth answers or labels are provided (often coming from the original dataset or obvious from context), so evaluation is well defined. For tasks that required new answer keys or classifications post-translation, the authors manually annotated them with consensus discussion to ensure accuracy. We did not compute a formal inter-annotator agreement due to the small team, but cross-checking was done to resolve any ambiguities.

---

### Task Format Non-Compliance Rate

The novelty of **Task Format Non-Compliance Rate** (TFNR) in TrustGen lies in how we use format violations within a trustworthiness benchmark. Instead of treating format adherence as a primary target for measuring instruction-following, TFNR acts as a structural control that flags answers whose form makes their semantic evaluation unreliable (e.g., mixed-format outputs, incomplete replies, refusals when refusal is not the object of study). These cases are separated from the main trust metrics so that we do not artificially inflate or deflate scores on dimensions such as ethics, fairness, or privacy when content correctness cannot be established with confidence. This design is intentionally different from utilitarian skill benchmarks: in capability evaluations, a format error is itself a failure, whereas in trustworthiness assessment a refusal or slight format deviation does not automatically make a model less truthful, fair, or ethical.

---

### Model Selection

Our primary goal was to compare Russian-adapted LLMs against their corresponding original multilingual counterparts on a unified trustworthiness benchmark. At the time of our experiments, all publicly available Russian-adapted LLMs that we could reliably evaluate had at most 72B parameters, and their base multilingual models have comparable scale. As a result, the majority of models in TrustGen belong to the small-to-medium regime; this is a direct reflection of the current Russian LLM ecosystem rather than an arbitrary restriction. We were additionally constrained by local compute resources, so most open-source models were run locally, while larger frontier models were evaluated via API.

---

### Meta-Review · Area_Chair_KjtJ · 2025-12-28

**Summary:**

This submission proposes TrustGen, a benchmark for evaluating the trustworthiness of large language models on Russian-language tasks across six dimensions. While all reviewers agree that addressing non-English trustworthiness evaluation, especially for Russian, is a meaningful direction, the consensus is that the paper’s contributions are limited. The benchmark largely reuses existing English datasets through translation and adaptation, adopts an existing trustworthiness taxonomy, and reports findings that mostly replicate well-known phenomena from prior English-centric studies. As a result, reviewers found the level of novelty and insight insufficient for ICLR, and the overall contribution was judged to be weak.

**Reviewer Concerns:**

The rebuttal partially addressed concerns about dataset construction and translation quality by clarifying the multi-stage annotation process and cultural adaptations, which alleviates some questions about basic validity. However, the core concerns remain largely outstanding. In particular, I am not convinced that TFNR constitutes a meaningful methodological innovation, nor that the benchmark provides new conceptual or empirical insights into trustworthiness beyond existing English benchmarks. Several reviewers also felt that the paper does not sufficiently analyze what is specifically Russian about the observed failures, beyond confirming known trends.

**Reviewer Scores:**

Revewers would likely maintain their reject scores, as their primary objections regarding novelty and contribution were not substantively changed by the rebuttal. Overall, the score distribution would remain firmly below the acceptance threshold, supporting a reject decision.

---

### Decision · Program_Chairs · 2026-01-26

Reject